# Rethinking Word-Level Auto-Completion in Computer-Aided Translation

**Xingyu Chen**[*‡], **Lemao Liu**[†§], **Guoping Huang,** [§]
**Zhirui Zhang**[§], **Mingming Yang**[§], **Shuming Shi**[§], **Rui Wang**[†‡]

[‡] Shanghai Jiao Tong University, China
[§] Tencent AI Lab, China
[‡] {galaxychen, wangrui12}@sjtu.edu.cn
[§] {redmondliu, donkeyhuang, jackzrzhang, shanemmyang, shumingshi}@tencent.com

## Abstract

Word-Level Auto-Completion (WLAC) plays a crucial role in Computer-Assisted Translation. It aims at providing word-level auto-completion suggestions for human translators. While previous studies have primarily focused on designing complex model architectures, this paper takes a different perspective by rethinking the fundamental question: what kind of words are good auto-completions? We introduce a measurable criterion to answer this question and discover that existing WLAC models often fail to meet this criterion. Building upon this observation, we propose an effective approach to enhance WLAC performance by promoting adherence to the criterion. Notably, the proposed approach is general and can be applied to various encoder-based architectures. Through extensive experiments, we demonstrate that our approach outperforms the top-performing system submitted to the WLAC shared tasks in WMT2022, while utilizing significantly smaller model sizes[¶].

## 1 Introduction

In recent years, more and more researchers have studied computer-aided translation (CAT) that aims to assist human translators to translate the input text (Alabau et al., 2014; Knowles and Koehn, 2016; Hokamp and Liu, 2017; Santy et al., 2019; Huang et al., 2021; Weng et al., 2019). The word-level auto-completion (WLAC) task (Casacuberta et al., 2022) is the core function of CAT, which involves predicting the word being typed by the translator given the translation context, as illustrated in Figure 1. Effective auto-completion has the potential to reduce keystrokes by at least $60\%$ during the translation process (Langlais et al., 2000). A

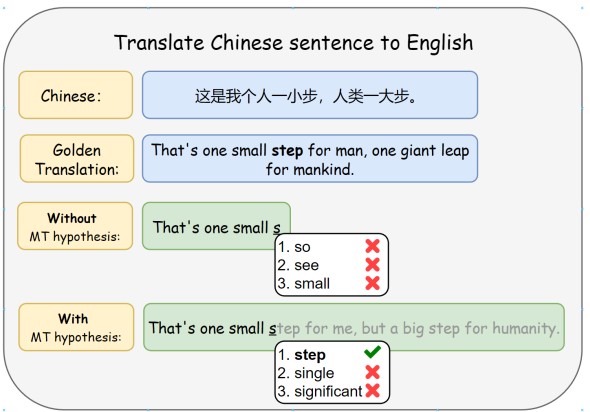

Figure 1: An example of word-level auto completion. Assume the human translator is going to input the *Golden Translation*. The auto-completion suggests the possible word candidates given the typed characters. It can be more accurate with the help of translation hypothesis from MT models.

user survey indicates that $90.2\%$ of participants find the word-level auto-suggestion feature helpful (Moslem et al., 2022). Therefore, WLAC plays an important role in CAT.

There are many existing methods for modeling WLAC, and they mainly differ in model architectures (Li et al., 2021; Yang et al., 2022b; Moslem et al., 2022; Yang et al., 2022a; Ailem et al., 2022). For example, Li et al. (2021); Yang et al. (2022a) design a BERT-like architecture to directly predict the target word while Yang et al. (2022b) employ a model similar to the auto-regressive NMT to predict the BPE tokens of the target word.

Take Figure 1 as an example, the task of WLAC can be described as: given the source sentence (*the Chinese sentence*), the partial translated sentence (*"That's one small"*) and the typed character sequence (*"s"*), predict the word **w** that the human translator is going to input (*"step"*). This paper goes beyond model architectures and reconsiders the essence of WLAC, posing a fundamental question (§3): what defines a correct word **w**? Theo-

---

[*]This work was completed during the Tencent AI Lab internship.

[†]The corresponding authors

[¶]Our code is publicly available at https://github.com/galaxyChen/WLAC-Joint-Training.

retically, a good **w** should appear in the reference translation, as illustrated in Figure 1. However, in practice, this criterion is not directly applicable for improving the performance of WLAC because the reference translation is unavailable during inference. Therefore, we attempt to relax this criterion by replacing the reference with the output from a trained machine translation system. The experimental results show that the relaxed criterion is reasonable: the predicted **w** that are present in the machine-translated hypothesis exhibit significantly higher accuracy compared to those absent in the hypothesis. Furthermore, we find that existing WLAC models usually fail to meet the relaxed criterion (see Table 2 later). This suggests an opportunity for improving their performance by meeting the criterion.

Based on the above finding, this paper presents a novel approach to enhance WLAC systems, which can be applied to different model architectures (§4). The key idea is to encourage WLAC's output to satisfy the relaxed criterion as much as possible, i.e., enhancing the agreement between WLAC and MT system such that the output **w** exists in the machine-translated hypothesis. Guided by this principle, two distinct methods are proposed. The first method achieves this agreement during the inference phase. This method requires running MT decoding during inference, which is inefficient. Moreover, its performance is limited by the quality of MT. To alleviate both issues, we introduce the second method, which achieves agreement through joint training of WLAC and MT. The intuition behind this method is that at the training stage, we generate the training data for the WLAC task from parallel corpus, so the WLAC's ground truth naturally exists in the reference translation. By jointly training the two models, we can implicitly learn the agreement between WLAC and MT. Notably, this method bypasses the need for MT decoding during inference, resulting in improved efficiency and removing the constraints imposed by MT's quality.

The effectiveness of our proposed method is validated through experiments on the four language directions of the WLAC shared task in WMT22 (§5). Our approach achieves substantial improvements across two distinct backbone models. Notably, it surpasses the performance of the WMT22's best system with only $20\%$ of the parameters and much simpler architecture.

This paper makes the following contributions:

- Rethinking the WLAC task by investigating the criterion for a golden WLAC prediction, and we find that the representative baseline models often fail to meet this criterion.

- To address this criterion violation, we propose an effective approach that leverages joint training to implicitly encourage agreement between WLAC and Machine Translation.

- Experimental results demonstrate that our proposed approach achieves remarkable improvements, surpassing state-of-the-art (SOTA) models by a significant margin. Notably, our approach also exhibits advantages in terms of model size and generality.

## 2 Backbone Models for WLAC

The WLAC task comes from a real translation scenario: a human translator is translating a source sentence, who has already translated part of the sentence, and is typing a new word. The input contains three parts: the source sentence **s**, the partial translation **c**, and the typed character sequence **t**. As shown in Figure 1, The WLAC task is to predict the word **w** that the translator is going to input (Li et al., 2021; Casacuberta et al., 2022),formulate as:

$$P(\mathbf{w}) = \mathcal{F}(\mathbf{s}, \mathbf{c}, \mathbf{t}), \qquad (1)$$

where $P(\mathbf{w})$ is the probability distribution of **w**, and $\mathcal{F}$ is the word prediction model. According toLi et al. (2021), the context can be categorized into four types depending on the input position, here we only consider the left context $\mathbf{c}_l$ and the right context $\mathbf{c}_r$ to the input, and both $\mathbf{c}_l$ and $\mathbf{c}_r$ could be empty.

In this section, we introduce two types of backbone models for the WLAC task. These backbone models serve as the foundations for our proposed techniques and experiments in the subsequent sections.

**Word-level Model** The first backbone is called All-In-One Encoder (AIOE), which adopts a BERT-like(Devlin et al., 2019) Transformer Encoder architecture for word prediction similar to Li et al. (2021). The AIOE takes the concatenation of the source sentence, context, and typed sequence as its input. The input format is: *s <sep> $c_l$ <tip> t <mask> $c_r$*, where $\mathbf{c}_l$ the left context to the input and $\mathbf{c}_r$ is the right context. Specifically, we append a *<mask>* token at the end of the typed character

sequence and leverage the final hidden state of the *<mask>* token for word prediction.

Despite its simplicity and efficiency, the AIOE model suffers from the out-of-vocabulary (OOV) problem, which can significantly hinder its performance. To this end, we introduce a variance of AIOE model that predicts word in sub-word level.

**Sub-word-level Model**  Extending the word-level AIOE model to sub-word-level is straightforward: we consider the task of predicting a sequence of sub-words as a generation problem, and introduce a Transformer Decoder to the AIOE model to perform the generation. We use Byte-Pair Encoding (BPE) (Sennrich et al., 2016) for the sub-word tokenization, and call this model *AIOE-BPE*. Yang et al. (2022b) also propose sub-word prediction at the BPE level, but their model treats it as a sequence of mask-prediction tasks, requiring interactively insert generated sub-words into the input. In comparison, the AIOE-BPE model offers a simpler architecture and has the ability to utilize advanced generation techniques such as beam search.

Due to the difficulty of labeling the WLAC data, we generate training data from parallel corpus for training the WLAC models, following the standard practice (Li et al., 2021; Casacuberta et al., 2022).

## 3  What Is a Good Prediction in WLAC?

In this paper, we rethink the WLAC task by posing a fundamental question: What is a "good" prediction given the source sentence $\mathbf{s}$, the context $\mathbf{c}$ and a typed character sequence $\mathbf{t}$? In the subsequent sections, we explore the criterion to assess a candidate predictions and use it to improve the performance during the inference.

### 3.1  A Criterion to Being a Good Prediction

**A principle criterion**  Rooted in the translation natural, the word $\mathbf{w}$ predicted by a WLAC model is a good prediction if $\mathbf{w}$ and the context $\mathbf{c}$ has the potential to lead to a high-quality translation $\mathbf{T}$ (*e.g.* the translation the human translator is going to input). In other words, $\mathbf{w}$ is likely to be a good prediction if it is contained in a high-quality (golden) translation $\mathbf{T}$. This can be considered as a principle criterion. Take Figure 1 as an example: the prediction *step* is good because it exists in the golden translation. However, this principled criterion is hard to apply in practice because we can not access the ground truth translation $\mathbf{T}$ during inference. Therefore, we explore a practical, relaxed

criterion next.

**A relaxed criterion**  Since the golden translation $\mathbf{T}$ is not available during inference, we propose to utilize the output of a machine translation model, named $\hat{\mathbf{T}}$, as the approximation of $\mathbf{T}$. Thus, we establish the relaxed criterion based on the $\hat{\mathbf{T}}$: $\mathbf{w}$ is a good prediction if $\mathbf{w}$ is contained in the machine-translated hypothesis $\hat{\mathbf{T}}$. In this case, we call $\mathbf{w}$ agrees with $\hat{\mathbf{T}}$ or the WLAC model agrees with the machine translation model. Given a data point $(\mathbf{s}_i, \mathbf{c}_i, \mathbf{t}_i, \mathbf{w}_i)$, we can compute the agreement by:

$$\hat{\mathbf{T}}_i = \mathcal{T}(\mathbf{s}_i, \mathbf{c}_i, \mathbf{t}_i), \qquad (2)$$

$$\text{Agreement}_i = \mathbb{I}(\mathbf{w}_i \in \hat{\mathbf{T}}_i), \qquad (3)$$

where $\mathcal{T}$ is a translation model and $\mathbb{I}(\cdot)$ is the indicator function. For the machine translation output $\hat{\mathbf{T}}$ may introduce noise and deviate from $\mathbf{T}$, it is natural to ask, how effective is this relaxed criterion? Therefore, we conduct preliminary experiments to justify the usefulness of this relaxed criterion.

### 3.2  Empirical Justification

To evaluate whether the relaxed criterion can be used as an indicator of good predictions, we examine the accuracy of words that agree (Agr. Acc.) and disagree (Disagr. Acc.) with the machine-translated outputs. We train an NMT model to generate top-5 translations for each source sentence in the test set, which serve as the noisy translation $\hat{\mathbf{T}}$. The analysis is conducted on the WMT22-WLAC testset for zh-en direction.

| WLAC | Agr. Acc. | Disagr. Acc. | $\Delta$ |
|---|---|---|---|
| AIOE | 63.75% | 44.97% | 18.78% |
| AIOE-BPE | 64.82% | 49.82% | 15.00% |

Table 1: Accuracy of agreement and disagreement under the relaxed criterion. Agr./Disagr. Acc. represents the percentage of correct predictions for the agreements/disagreements between WLAC and MT model.

**Is the relaxed criterion reasonable?**  Table 1 shows the Arg. Acc. and Disagr. Acc for two backbone models, which reveals insightful findings. Although the agreement accuracy (about 64%) is limited by the discrepancy between noisy and golden translation, the accuracy drops dramatically if the prediction disagrees with the translation $\hat{\mathbf{T}}$. Therefore, despite the presence of noisy translations, these results support the relaxed criterion to some extent: predictions that agree with translation

are more likely to be correct, whereas violations of the relaxed criterion (disagreements) would lead to poor accuracy.

| WLAC | Agreement | Disagreement |
|------|-----------|--------------|
| AIOE | 47.36% | 52.64% |
| AIOE-BPE | 49.03% | 50.97% |

Table 2: The percentage of agreement between NMT and each backbone WLAC model.

**Backbone models usually fail to meet the relaxed criterion.** Additionally, we analyze the degree of agreement between each backbone model with MT and the result is illustrated in Table 2. Surprisingly, we observe that less than a half of the predictions agree with the translations, i.e., WLAC model violates the relaxed criterion with a large probability. Considering the substantial accuracy gap between agreement and disagreement, there is significant room for performance enhancement.

### 3.3 The relation between *accuracy* and *agreement*

One might wonder why we want to introduce a new criterion to assess the prediction, in addition to the original accuracy metric. The primary distinction between accuracy and agreement lies in their roles within the WLAC task. Accuracy, being the official metric, directly evaluates the correctness of label predictions. In contrast, agreement operates independently of labels, quantifying the presence of prediction candidates in translation hypotheses. While accuracy assesses "correctness", it can't provide insights for model optimization. In the subsequent section, we describe how to leverage agreement to enhance WLAC models.

## 4 Enhancing WLAC with MT by Agreement

Motivated by the findings from Section 3, we propose two different approaches which *improve the agreement between WLAC and machine translation* for overall improvements in WLAC performance.

### 4.1 Enhancing agreement via Joint Inference

One approach to enhance agreement is jointly consider the WLAC predictions and machine translation results during inference. We begin by generating the top-k predictions from the WLAC model. Then, we use a MT model to generate translation hypothesis based on the source sentence. Next, we examine each word in the predictions and check if it is included in the translation. The first word in the top-k list that exists in the translation is selected as the final prediction. This strategy manually align the prediction with translation in a flexible way: the choice of WLAC model and translation model is arbitrary. The final performance is closely related to the choices of models.

However, this approach heavily relies on the quality of translation. A preliminary analysis show that for a naive MT model, only $44.6\%$ of the WLAC labels exist in the translation. In such scenarios, enhancing agreement alone does not guarantee performance improvements. One possible solution is to enhance the input of MT model. We propose a *Context MT* model, which takes additional translation context and typed sequence as input, and generates full target sentence. The input of *Context MT* is the same as WLAC, so it's a better approximation of the golden translation model.

### 4.2 Enhancing agreement via Joint Training

One drawback of joint inference method is that the WLAC model isn't aware of the translation task during training, which means that the top-k predictions may deviate from the ground truth. To overcome this limitation, we propose a joint training approach, wherein the WLAC model and the MT model are trained together using a shared backbone encoder. Specifically, we extend the backbone model by introducing an MT decoder, transforming the backbone model into an MT model. Here the MT model is the same as *Context MT* model described in §4.1. We define the training loss of the joint training model as the combination of the WLAC loss and the translation loss, represented as follows:

$$\mathcal{L} = \alpha \cdot L_{\text{WLAC}} + (1 - \alpha) \cdot L_{\text{MT}}, \qquad (4)$$

where $\alpha$ is a hyper-parameter controlling the balance between the two losses. To enhance the interaction between two tasks, we also share the final word prediction layer between the backbone model and the decoder. As described in section 5.1, the training data of WLAC is generated from parallel corpus, so there will be a full agreement between WLAC label and ground truth translation at the training stage. This agreement enables the WLAC model to learn how to accurately predict words within the translations. Besides, the MT

model can learn to generate translations based on the context provided by the WLAC predictions. By jointly training the two models, we enable them to mutually benefit from each other's knowledge and improve their respective tasks.

The key advantage of joint training is that once the training is completed, we can only keep the backbone model and discard the MT decoder. Note that the backbone encoder can receive optimization signals from both the WLAC task and the translation task, so the backbone model has acquired the skill to agree with translation during training process. This enables us to maintain the agreement capabilities while preserving a small and efficient inference model.

## 5 Experiment

### 5.1 Datasets

We conduct experiments on two language pairs: English-Chinese and English-German. The zh-en dataset we used is the UN Parallel Corpus V1.0 from WMT17. For en-de, we use the training data from WMT14. We adopt the following strategy on parallel sentences to generate WLAC training data[*]: firstly, we sample a target word $\mathbf{w}$ from the target language sentence, then we sample spans respectively from the left and right context of the target word, denoted as $\mathbf{c}_l$ and $\mathbf{c}_r$. Additionally, we sample a typed sequence from the target word. To sample typed sequence from Chinese words we use the pypinyin[†] tool. All models are trained on the generated training data, with WMT21 data serving as the validation set. For evaluation, we utilize the test set from the WMT22 WLAC shared task.

### 5.2 Models for Comparison

**GWLAN** Proposed by Li et al. (2021), GWLAN model is word-level model with two encoders.

**HW-TSC** The models proposed by Yang et al. (2022b) are the winners of the WMT22 WLAC shared task across three language directions. It's a BPE-level model with encoder-decoder architecture.

**AIOE** This model is the word-level model described in section 2. The AIOE model contains 6 Transformer Encoder layers. See appendix B for more training details.

---

[*]https://github.com/lemaoliu/WLAC/blob/main/scripts/generate_samples.py
[†]https://github.com/mozillazg/python-pinyin

**AIOE-BPE** This is the sub-word-level model described in section 2. The BPE decoder is a 6 layer Transformer Decoder.

**AIOE-Joint** The AIOE model with machine translation joint training. The MT decoder contains 6 Transformer Decoder layers.

**AIOE-BPE-Joint** The AIOE-BPE model with machine translation joint training. We share the final word prediction layer of MT decoder and AIOE-BPE backbone decoder.

### 5.3 Comparison among Agreement Methods

We firstly compare the performance of joint inference method and joint training method. For joint inference method, we use the word-level backbone AIOE model for the WLAC model, and consider two kinds of machine translation model: translation model trained on parallel corpus (MT) and translation model trained on WLAC input and translation output (*Context MT*). We also consider the Google Translator(GT)[‡] to see the impact of translation quality. For the joint training method, we use AIOE-Joint model. All the experiments are conduct in zh-en direction. The result is reported in Table 3.

| Method | Acc. | Agr. | Agr. Acc. |
|--------|------|------|-----------|
| AIOE | 53.87 | 47.36% | 63.75% |
| AIOE+MT | 54.20 | 58.61% | 57.70% |
| AIOE+CMT | 56.01 | 74.51% | 61.98% |
| AIOE+GT | 56.16 | 44.51% | 62.50% |
| AIOE+JT | 59.75 | 65.50% | 67.51% |

Table 3: Comparison of joint-methods. *Acc.* is the accuracy of WLAC task. *Agr.* is the percentage of agreements. *Agr. Acc.* is the percentage of the accurate prediction among agreements.

It is observed that joint inference with MT and CMT models significantly improve the agreement, although the percentage of accurate agreement decreases when combined with the translation models. This discrepancy can be attributed to the fact that the machine translation models are unaware of the WLAC task, and their translations may deviate from the ground truth. Consequently, enhancing agreement does not guarantee an improvement in the quality of WLAC predictions under this situation. On the other hand, using a better MT model

---

[‡]https://cloud.google.com/translate/

(Google Translator here) can bring better performance than using NMT system or Context MT system, but the agreement drops greatly. We attribute this phenomenon to two primary factors: 1. we only use top 1 translation from Google Translator while we use top 5 translation from MT/CMT model. 2. The high agreement accuracy suggests that Google Translator can closely approximate human translators. As a consequence, it is plausible that the performance of the foundational AIOE model may serve as the limiting factor in this context. From the perspective of overall accuracy, although the joint inference method outperforms the backbone AIOE model, there is still a large gap compared to the joint training method. Based on these findings, we only focus on the joint training method for the subsequent experiments.

### 5.4 Main Results

The evaluation result on tthe WMT22 WLAC shared task is reported on Table 4. Compared to HW-TSC, our word-level methods have obtained better performance on zh-en and de-en. One exception is en-de, the word-level model performed badly because it suffers from OOV problem, where about $17\%$ labels are OOV. After replacing the backbone with BPE-level model, our method show superior performance compared to the SOTA in all directions. No matter which backbone is used, our joint training method can greatly improve the backbone performance, indicating that our method is a general framework and has the potential to be applied to more encoder based models. Another obvious advantage of our model is its superior parameter efficiency. Despite being only $15\%$ of the size of the HW-TSC model during inference, our best AIOE-BPE-Joint model outperforms the HW-TSC model significantly.

### 5.5 Ablation Study

**Translation performance** Our method has demonstrated the importance of leveraging machine translation knowledge to solve the WLAC task, but is it possible to solve the task directly using a translation model? We use the same parallel training data to train a translation model and compare the performance of different inference strategy: 1) *Upper Bound:* if the top-k generated translations contain the target word, we treat it as "correct". 2) *Prefix Match:* We use the typed sequence to match the top-k translations and take the most common matched word as the prediction.

We employ the *Context MT* model, as described in Section 4.1, to generate the top-5 translations using beam search. Then we apply the aforementioned inference strategies to these translations. We also evaluate the performance of GPT-*text-davinci-003*[§](we set the temperature as 0), including directly solving the task and matching from the translation. The prompts used in our experiments are provided in Appendix A. The experimental results for the zh-en language pair are presented in Table 5. Notably, while the translation upper bound is comparable to our joint training method, there still exists a large gap between the matching method and the upper bound. Considering that the inputs of *Context MT* and AIOE model are the same, the huge difference in their performance indicates that our joint training method is more effective in utilizing translation knowledge to solve WLAC task. The GPT performs badly in the WLAC task, indicating that we need to specifically optimize the model for the WLAC task.

**The impact of MT task** The influence of the hyper-parameter $\alpha$ on the model performance, as outlined in equation 4, directly reflects the impact of translation task. By setting $\alpha$ to $0$, the model is essentially a translation model with additional context input. If $\alpha = 1$, the model corresponds to the AIOE model without joint training. In Figure 2, we present the accuracy achieved at varying values of $\alpha$. Notably, as $\alpha$ increases from $0$ to $0.75$, the accuracy increases rapidly. This observation highlights the difference between the translation task and the WLAC task, emphasizing the necessity of optimizing the model specifically for the WLAC task to achieve better performance. Interestingly, even with $\alpha$ set to $0.99$, the performance remains comparable to the best achieved performance. This finding is remarkable, as it suggests that even a small signal from the translation task can greatly enhance the WLAC task's performance when compared to the model with $\alpha$ set to $1$. Consequently, our proposed joint training method effectively integrates the translation task into the WLAC task, resulting in substantial improvements.

### 6 Analysis & Discussion

The experiment results have demonstrated the effectiveness of our method, but how does the MT task help WLAC? In this section, we conduct de-

---

[§](https://platform.openai.com/docs/guides/gpt)

| Model | #Parameters | zh-en | en-zh | en-de | de-en |
|---|---|---|---|---|---|
| GWLAN | 105M | 51.11 | 48.90 | 40.69 | 53.87 |
| HW-TSC | 526M | 59.40 | - | 63.82 | 62.06 |
| AIOE | 80M | 54.13 | 52.88 | 39.43 | 56.40 |
| AIOE-BPE | 74M | 57.17 | 52.97 | 56.83 | 61.62 |
| AIOE-Joint | 80M(105M) | 59.75 | 56.59 | 44.67 | 62.77 |
| AIOE-BPE-Joint | 74M(100M) | **61.08** | **58.09** | **64.59** | **66.91** |

Table 4: Experiment results on WMT22 WLAC test set. Results are reported as accuracy. The number of parameters in brackets means parameters in training stage.

| Method | Accuracy |
|---|---|
| UpperBound | 58.05 |
| PrefixMatch | 45.69 |
| GPT-UpperBound | 42.33 |
| GPT-Direct | 17.89 |
| AIOE | 53.87 |
| AIOE+Joint Training | **59.75** |

Table 5: Zh-En results for translation-only methods.

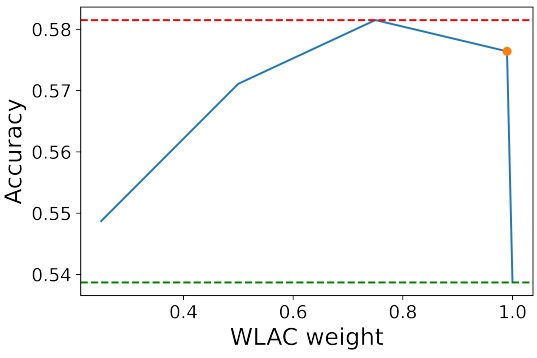

Figure 2: The impact of different $\alpha$ on the AIOE accuracy. Red dashed line is the best performance and the green represents the accuracy without joint training.

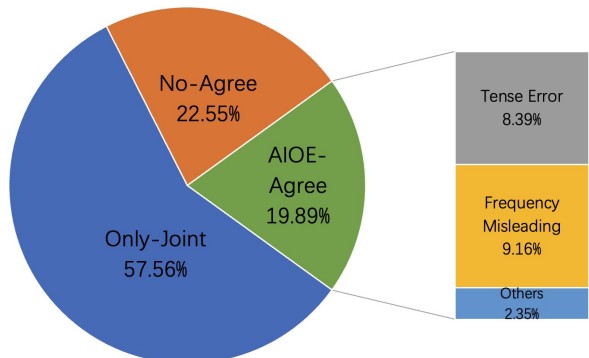

Figure 3: Improvement analysis on cases where AIOE prediction is wrong while AIOE-Joint is right. *AIOE-Agree*: the AIOE model agrees with the translation. *Only-Joint*: only the AIOE-Joint model agrees with the translation. *No-Agree*: none of the models agrees with the translation.

tailed analysis on the influence of the translation task and study the relation between generated translations and predicted words. All the experiments are based on word-level AIOE-Joint model in zh-en direction.

## 6.1 Improvement Analysis

To gain insights into the contributions of the translation task to WLAC prediction, we conduct a detailed analysis of the improvements observed after introducing joint training. Let $\mathbf{w}_e$ represent the predictions of the AIOE model, $\mathbf{w}_m$ denote the predictions of the AIOE-Joint model, $\mathbf{w}$ denote the ground truth, and $\hat{\mathbf{T}}$ represent the translation generated by the AIOE-Joint model. We focus our

analysis on cases where $\mathbf{w}_e \neq \mathbf{w}_m$ and $\mathbf{w}_m = \mathbf{w}$. Based on whether $\mathbf{w}_e$ and $\mathbf{w}_m$ agree with $\hat{\mathbf{T}}$, we define three groups: *AIOE-Agree*, *Only-Joint* and *No-Agree*. The percentage of each group is illustrated in Figure 3. Next, we will conduct detailed analysis on each group.

**Does the translation help the WLAC prediction?** The *Only-Joint* group contains the cases that only AIOE-Joint model agrees with translation, which is the most notable part of improvement. The translation process can be viewed as a planning procedure, guiding word prediction by leveraging knowledge of the complete translation. In this scenario, the model simply needs to select the most likely word from the translation. Though we do not explicitly generate translation during inference, the backbone model of AIOE-Joint has learnt how to encode the input for the MT task, so the translation implicitly helps the model to make correct prediction.

**Why AIOE model fails when it agrees with translation?** From Figure 3, there are about 22.55% of the cases where the AIOE model agrees with

the translation. However, despite the agreement, the model still fails to make the correct prediction. Further analysis of this group reveals two types of errors: tense errors and frequency misleading. Tense errors occur when the AIOE prediction and the label correspond to different tenses of the same word. We utilize the nltk.stem tool [¶] to identify tense errors. Interestingly, the introduction of MT joint training resolves the tense errors in this group, indicating that the model can generate morphologically accurate predictions under translation context. Frequency misleading refers to the phenomenon where the AIOE model tends to predict high-frequency words. We compute the percentage of cases where AIOE predicted words are more frequent than AIOE-Joint predictions. Remarkably, approximately half of the cases in the AIOE-Agree group fall under this error category. This suggests that, with the assistance of translation, the WLAC model can prioritize words that are contextually appropriate rather than relying on repetitive high-frequency words.

**Does the AIOE-Joint model "copy" words from translation?** The *No-Agree* group offers a new insight about how the joint model works: a substantial portion of correct predictions is not present in the translation, suggesting that the word prediction module does not merely copy words from the translations. Instead, it utilizes translations as references to achieve the most accurate predictions.

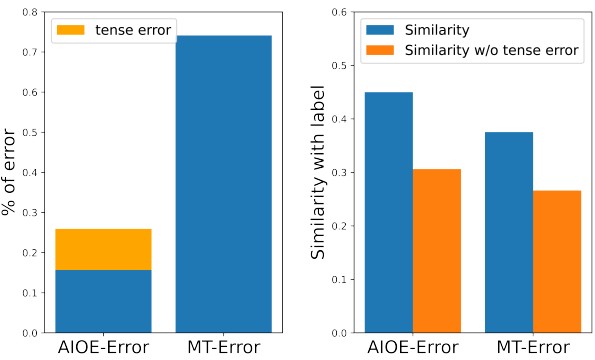

Figure 4: Error Analysis. *AIOE-Error*: WLAC label exists in the translation, *MT-Error*: translation does not the contain WLAC label.

## 6.2 Error Analysis

Lastly, we dive into the cases where the AIOE-Joint model encounters failures. In this analysis, we consider the AIOE-Joint prediction $\mathbf{w}_m$, the

ground truth $\mathbf{w}$, and the translation $\hat{\mathbf{T}}$ generated by the AIOE-Joint model. We classify the error cases based on whether the translation $\hat{\mathbf{T}}$ includes the word $\mathbf{w}_m$ and $\mathbf{w}$. Additionally, we report the cosine similarity between $\mathbf{w}_m$ and $\mathbf{w}$ using the fastText tool[‖]. The results are illustrated in Figure 4.

**MT failure** The majority of errors can be attributed to the *MT-Error* group, where the labels are absent in the translation. The low similarity between the labels and predictions in this group indicates that the AIOE-Joint model fails to capture essential information from the input under such circumstances. The significant divergence between the model's translation and the golden translation is a key factor contributing to this failure. We attribute the translation fails in these cases to the lack of sufficient context. In Table 6, we present the average length of context and the percentage of zero-context cases for different groups. Notably, the *MT-Error* group exhibits the lowest average context length and the highest percentage of zero-context cases. These findings strongly suggest that the model struggles to generate reasonable translations when there is limited context. This issue could potentially be addressed by employing more advanced sampling techniques or using a more powerful MT decoder. We will leave it as future work.

| Group | Avg. Context | Zero-Context% |
|---|---|---|
| Err. | 5.63 | 26.67% |
| AIOE-Err. | 6.00 | 22.82% |
| MT-Err. | 5.50 | 28.02% |

Table 6: Statistics about context length. Group *Err.* contains all the AIOE-Joint failures. Group *AIOE-Err.* and Group *MT-Err.* represents *AIOE-Error* and *MT-Error* respectively. *Avg. Context* is the averaged length of context, and *Zero-Context%* is the percentage of zero-context cases.

**Backbone failure** The *AIOE-Error* group indicates that the model may deviate during word prediction and struggle to identify the correct target from the translation. However, the similarity of this group is unexpectedly high. To figure out why AIOE model fails in these cases, we look into specific error types of this group. Further analysis reveals that there are approximately 20% tense errors. After eliminating the tense errors from this group, the similarity decreases dramatically, sug-

---

[¶]https://www.nltk.org/api/nltk.stem.html

[‖]https://fasttext.cc/

gesting that the model fails to capture crucial information from the input under this situation. These errors could be mitigated by enhancing the encoder. Through our comprehensive analysis, a significant advantage of our proposed method emerges: the model's behavior is more interpretable, enabling easier identification of bottlenecks.

# 7 Related Work

## 7.1 Interactive Machine Translation

Interactive machine translation is a sub-field of computer aided translation, where human translator would write the translation interactively. An IMT system would generate a translation hypothesis based on the current context, and the human translator would be asked to post edit the sentence. Each time the translator edit the sentence, a new hypothesis would be generated, and this process would continue until the translator is satisfied with the translation. Different IMT systems have emerged (Green et al., 2014; Hokamp and Liu, 2017; Weng et al., 2019; Wang et al., 2020; Huang et al., 2021), and it's reported that IMT systems can improve translation quality (Green et al., 2014) and is benefit to the potential users (Casacuberta et al., 2009; Barrachina et al., 2009).

## 7.2 Auto Completion

Auto completion is the core function of IMT, including sentence-level auto completion and word-level auto completion. Previous work mainly focus on sentence-level auto completion (Alabau et al., 2014; Zhao et al., 2020), where the system would complete the whole translation based on user typed prefix. However, sentence-level auto completion requires to generate a complete sentence every time the user inputs, and the overhead of the system is very high.

Word-level auto completion only predicts the word that the user is going to input. Previous studies have explored predicting the next word based on the prefix and typed chars (Langlais et al., 2000; Santy et al., 2019), while our work focuses on real-world translation scenarios like post-editing (Vasconcellos and León, 1985; Green et al., 2014). Yang et al. (2022a) proposed to use encoder-like architecture to predict the target word and Yang et al. (2023) designed an energy based model for reranking, Yang et al. (2022b) additionally introduce machine translation as pre-training task, and Moslem et al. (2022) examined the performance of pure translation methods without training on WLAC task. We emphasize the importance of employing translation knowledge during the WLAC training, so we proposed to jointly train the two tasks, incorporating translation into word prediction by a soft yet effective way.

# 8 Conclusion

This paper firstly rethinks the essence of the "correct" prediction for WLAC and we propose *agreement*, a measurable criterion to judge whether a prediction word is good or not. Surprisingly, we find that existing WLAC models usually violate this criterion with a high probability. Based on this findings, we present two effective strategies: joint inference and joint training with machine translation (MT), aimed at enhancing WLAC performance by encouraging the model to adhere to the agreement criterion. Extensive experiments show that the proposed approach surpasses the best system submitted to the WLAC shared tasks in WMT2022, with much smaller model size. Additionally, our result analysis highlights the effective integration of MT knowledge into the WLAC task, resulting in improved performance. The error analysis also reveals the directions for further optimization, which we will leave as future work.

# Acknowledgements

Xingyu and Rui are with the MT-Lab, Department of Computer Science and Engineering, School of Electronic Information and Electrical Engineering, and also with the MoE Key Lab of Artificial Intelligence, AI Institute, Shanghai Jiao Tong University, Shanghai 200204, China. Rui is supported by the General Program of the National Natural Science Foundation of China (62176153), the Shanghai Pujiang Program (21PJ1406800), the Shanghai Municipal Science and Technology Major Project (2021SHZDZX0102), the Alibaba-AIR Program (22088682), and the Tencent AI Lab RBFR2023012.

# Limitations

Based on our error analysis, a significant portion of errors can still be attributed to MT errors, indicating that even with joint training, there are discrepancies between the MT model and the golden translation.

Furthermore, our analytical experiments were conducted specifically on the zh-en language pair,

and the generalizability of our findings to other languages may vary. While our main experiments have demonstrated the effectiveness of the proposed joint training method, it is essential to examine the improvement analysis and error analysis for specific linguistic phenomena, such as the prediction of German grammatical gender. To gain a comprehensive understanding of the effectiveness of the joint training approach, our future work includes extensive experiments on multiple languages.

Lastly, the current WLAC task primarily focuses on high-resource languages. Exploring the performance of our method in low-resource scenarios is also an important area for future investigation.

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

## A  GPT prompts

### A.1  Prompts for directly solving problems

We design different prompts for different types of context as described in (Li et al., 2021). *Words in italic* will be replaced by the actual input for each case.

**Prefix**  A human translator is translating a Chinese sentence into English. The Chinese sentence is "*source sentence*". The translator has already translated part of the sentence, the left context is "*left context*". The translator is now typing a new word after the left context, the typed sequence is "*typed sequence*", what's the word the translator is going to input? Your answer should only contain the word.

**Suffix**  A human translator is translating a Chinese sentence into English. The Chinese sentence is "*source sentence*". The translator has already translated part of the sentence, the right context is "*right context*". The translator is now typing a new word before the right context, the typed sequence is "*typed sequence*", what's the word the translator is going to input? Your answer should only contain the word.

**Bi-context**  A human translator is translating a Chinese sentence into English. The Chinese sentence is "*source sentence*". The translator has already translated part of the sentence, the left context is "*left context*", and the right context is "*right context*". The translator is now typing a new word between the left context and right context, the typed sequence is "*typed sequence*", what's the word the translator is going to input? Your answer should only contain the word.

**Zero-context**  A human translator is translating a Chinese sentence into English. The Chinese sentence is "*source sentence*". The translator is now typing a word, the typed sequence is "*typed sequence*", what's the word the translator is going to input? Your answer should only contain the word.

## B  Training Details

For all AIOE model, we use a Transformer Encoder for 6 layers. The embedding size is 512, the dimension for feed-forward layer is 2048. Each layer has 8 attention heads. For AIOE-BPE model, we additionally add a Transformer Decoder with 6 layers.

For AIOE model, we use a joint-vocabulary with the size of 120000. For AIOE-BPE model, the vocabulary size is 66630 for English-Chinese pair and 59918 for English-German pair.

The learning rate for training is 5e-4. We optimize the model for 200000 steps with a batch size of 32000 tokens.