# OpenReview forum: "Rethinking Word-Level Auto-Completion in Computer-Aided Translation"
_EMNLP/2023/Conference — EMNLP 2023 Main_

### Official Review · Reviewer_Nd7e · 2023-08-04

**Soundness:** 4

**Excitement:**

4: Strong: This paper deepens the understanding of some phenomenon or lowers the barriers to an existing research direction.

**Missing References:**

N/A

**Paper Topic And Main Contributions:**

By rethinking the definition of a good WLAC prediction, this paper introduces a criterion that a better prediction is more likely to be contained in a high-quality translation. Then this paper proposes to jointly train WLAC and MT models to encourage agreement between the predicted words of WLAC and the translation from MT models. AIOE and its variants outperform previous SOTA baselines on the WMT22 WLAC test set, demonstrating the effectiveness of the proposed method. The intuition and motivation, method introduction, experimental setups, and detailed result analysis are well elaborated, making the work sound and reproducible.

**Questions For The Authors:**

Line 537-539: "We attribute the translation fails in these cases to the lack of sufficient context."
Translation error is common since the MT model can not be perfect, as also mentioned in Limitations. Hence it is suggested to provide evaluation metrics, such as BLEU, of the Transformer-based MT model on the four translation directions in Table 3. This will help readers better understand the basic translation abilities of the jointly trained MT model.

**Reasons To Accept:**

1. This paper proposes an effective method to handle WLAC tasks with the help of MT models. The proposed method performs effectively on the WMT22 WLAC test set, surpassing previous SOTA baseline models.
2. The intuition and motivation, method introduction, experimental setups, and detailed result analysis are well elaborated, making the work sound and reproducible.

**Reasons To Reject:**

Code releasing is encouraged for better reproducibility.

**Reproducibility:**

4: Could mostly reproduce the results, but there may be some variation because of sample variance or minor variations in their interpretation of the protocol or method.

**Reviewer Confidence:**

4: Quite sure. I tried to check the important points carefully. It's unlikely, though conceivable, that I missed something that should affect my ratings.

**Typos Grammar Style And Presentation Improvements:**

Presentation Improvements:
1. Table 4 appears in Section 5.3, which precedes Table 3 in Section 5.4. It would be better to adjust the table format and reorder them.
2. Line 411: A citation or URL link is needed when mentioning "GPT-text-davinci-003". Besides, the generation temperature of the GPT model should be provided.
3. Figure 3: The font style and size should be improved for better presentation.

---

> ### Author Rebuttal · Authors · 2023-08-28
>
> Thank you for the insightful comments. We will release our reproducible code along with the next version of paper. We address your comments below.
>
> **Q1 Provide evaluation metrics, such as BLEU, of the Transformer-based MT model on the four translation directions in Table 3**
>
> Thank you for point out this problem. We understand that a translation metric such as BLEU can provide better insights about how the MT model fails. However, the test set provided by WMT WLAC shared task does not provide a golden translation, so we can't compute the BLEU score on it. We will try to report the BLEU results on our synthetic dataset in the next version of paper.
>
> **Typos Grammar Style And Presentation Improvements**
>
> Thank you for the detailed suggestions. We will adjust the location of tables and improve figure quality for better presentation. The temperature of GPT-text-davinci-003 is 0, we will also provide it in the next version.

---

### Official Review · Reviewer_4EMT · 2023-08-05

**Soundness:** 4

**Excitement:**

4: Strong: This paper deepens the understanding of some phenomenon or lowers the barriers to an existing research direction.

**Paper Topic And Main Contributions:**

This paper addresses the problem of word-level auto-completion (WLAC), where the model tries to complete a partially typed word sequence $\mathbf{t}$ into a full word $w$, given the source sentence $\mathbf{s}$ and target context $\mathbf{c}$.

The first main contribution of the paper is a preliminary analysis that shows that MT output is also a reasonable proxy to estimate WLAC models' performance, and existing models (AIOE) don't work well when evaluated with this proxy. This paves the way for the other contribution, which are two different improvements: (1) at inference time, use MT output to help select a good completion among the top-k list predicted by the model; (2) jointly train the model with MT capabilities, so the model can predict something that will likely be in the MT output.

Experiments show that the proposed method is very effective -- on the WMT22 WLAC benchmark, it beats the SToA model with only 15% of its parameter size.

**Questions For The Authors:**

Out of curiosity: Do you have an idea how much MT quality affects the performance of the AIOE+MT model? For example if you use Google Translate or GPT-4, can you do better than AIOE-Joint?

**Reasons To Accept:**

1. The proposal is very well-motivated by the preliminary analysis.
2. Experiment results are strong.
3. The error analysis and subsequent discussion give good insights into where the model improves and where it still falls short.

**Reasons To Reject:**

My main complaint with this paper is its presentation quality. A lot of concepts could be described more clearly to avoid confusing the readers, especially those who are not familiar with the WLAC task. I will elaborate on my feedback in the presentation improvement box.

In my opinion, this is one of these cases where the "required edit" option in journals would have been useful but is not an option for EMNLP. I think the content is good enough so I would rather not block this paper for this reason but I hope the authors do address my comments before camera-ready.

**Reproducibility:**

4: Could mostly reproduce the results, but there may be some variation because of sample variance or minor variations in their interpretation of the protocol or method.

**Reviewer Confidence:**

3: Pretty sure, but there's a chance I missed something. Although I have a good feel for this area in general, I did not carefully check the paper's details, e.g., the math, experimental design, or novelty.

**Typos Grammar Style And Presentation Improvements:**

I think I had two big confusions when I read the paper that took me a while to recover from them:

- You are not proposing a new metric (which I misunderstood from the word choice "criterion"), but Section 3 mainly serves as a preliminary analysis that motivates the subsequent model improvements. If you are indeed proposing a metric, you will need to do a very different set of evaluations here.
- The notion of "accuracy" is what you compute when comparing with the ground truth, as in the test set, while "agreement" is checking if the word exists in the translated sentence. This is used extensively throughout the paper, so I think it's worth to be explained more explicitly.

Other than that, a few more small things:

- L56: I would say during "inference" instead of during "testing" here
- L120: typed *character* sequence $\mathbf{t}$
- L225-234: I would add how much ground truth agrees & disagrees with MT output for comparison in Table 2, in order to strengthen your argument there
- L257: This is talking about WLAC *gold* labels, right? If yes this is what you want to include in Table 2.
- L483: AIOR -> AIOE
- Figure 4: I find the lines confusing. Probably remove them and use some larger markers in the figure (e.g. cross or triangle).

---

> ### Author Rebuttal · Authors · 2023-08-28
>
> Thank you for the insightful comments. We admit that some concepts in our paper should be clarified and we will improve it in the next version. We address your comments below.
>
> **Q1 Do you have an idea how much MT quality affects the performance of the AIOE+MT model? For example if you use Google Translate or GPT-4, can you do better than AIOE-Joint?**
>
> The MT quality does affects the performance of AIOE+MT model. We use the Google Translation API to generate translation hypothese and apply joint inference on it. The result is shown below:
>
> | Model                  | Acc.  |
> | ---------------------- | ----- |
> | AIOE+MT                | 54.20 |
> | AIOE+CMT（Context MT） | 56.01 |
> | AIOE+Google Translator | 56.27 |
> | AIOE+Joint Training    | 59.75 |
>
> As we can see, the performance of the joint inference method is influenced by the quality of MT model. Using a better MT model  (Google Translator here) can bring better performance than using our NMT systems or Context MT system, but the performance still lacks behind the joint training method. This results highlight the importance of incorporating translation signal into the WLAC model.
>
> **Q2 Are you proposing a new metric?**
>
> No, the proposed criterion "agreement" is *not* a metric for the WLAC task, because "agreement" doesn't access the labels of WLAC task, it just computes how many predictions exist in the translation hypotheses.  The criterion "agreement" reveals the relations between the WLAC task and the MT task, which serves as a preliminary analysis that motivates the subsequent model improvements.
>
> **Q3 The relationship between "accuracy" and "agreement"**
>
> Yes, your understanding of "accuracy" and "agreement" is correct. We admit that these two concepts are not well described in the current version of paper, so we will further clarify them in the next version. Thank you for pointing out this problem.
>
> **Typos Grammar Style And Presentation Improvements**
>
> Thanks for your detailed comments, we will fix the typos and improve the presentation according to your suggestions. We will re-organize the Table 2 to include more information. We will also update the Figure 4 to make it less confusing in the next version.

---

### Official Review · Reviewer_1XZn · 2023-08-05

**Soundness:** 4

**Excitement:**

4: Strong: This paper deepens the understanding of some phenomenon or lowers the barriers to an existing research direction.

**Paper Topic And Main Contributions:**

This paper revisits the problem of word-level auto-completion by studying its relation to machine translation. It analyzes the limitations of previous methods and proposes joint inference and joint training of WLAC and MT. The proposed method outperforms previous approaches, and further analysis also support the paper.

**Questions For The Authors:**

- Is it possible to combine joint training and joint inference? If so, does it result in performance improvements?
- Can you provide more details or codes for generating the synthetic dataset? While it is well-written, some minor details are required for reproduction (e.g., span sampling probability, ratio of MT and WLAC data, and dynamic/static sampling of synthetic data)

**Reasons To Accept:**

- The method is well-motivated, simple, and the results are promising.
- The experiments and analysis provided are sufficient to support the effectiveness of the method.

**Reasons To Reject:**

- I don't see any clear reason to reject this paper.

**Reproducibility:**

3: Could reproduce the results with some difficulty. The settings of parameters are underspecified or subjectively determined; the training/evaluation data are not widely available.

**Reviewer Confidence:**

3: Pretty sure, but there's a chance I missed something. Although I have a good feel for this area in general, I did not carefully check the paper's details, e.g., the math, experimental design, or novelty.

---

> ### Author Rebuttal · Authors · 2023-08-28
>
> Thank you for the insightful comments. We address your comments below.
>
> **Q1. Is it possible to combine joint training and joint inference? If so, does it result in performance improvements?**
>
> Yes, it's possible to combine these two methods. We show the result of zh-en below:
>
> | Model                               | Acc.  |
> | ----------------------------------- | ----- |
> | AIOE                                | 53.87 |
> | AIOE+Joint Inference                | 56.01 |
> | AIOE+Joint Training                 | 59.75 |
> | AIOE+Joint Training+Joint Inference | 59.28 |
>
> As we can see, the performance of combining two methods is comparable with using joint training only. The procedure of joint inference is: 1. Use MT model to generate translation hypotheses. 2. Check whether the top-5 predictions of WLAC model exist in the translation hypotheses, if so, use it as the final prediction. The performance of joint inference is greatly hindered by the quality of MT model. Even the joint training method can produce more accurate prediction candidates, the poor MT hypotheses would mislead the prediction selection.
>
> **Q2. Can you provide more details or codes for generating the synthetic dataset? While it is well-written, some minor details are required for reproduction (e.g., span sampling probability, ratio of MT and WLAC data, and dynamic/static sampling of synthetic data)**
>
> Yes, we will release our reproducible code along with the next version of paper. We follow the official WLAC data generation script (https://github.com/lemaoliu/WLAC/blob/main/scripts/generate_samples.py) to generate training data for the WLAC task. The ratio of translation data and the WLAC data is 1:1, and we dynamically generate training examples at the training time.

---

### Meta-Review · Area_Chair_GkSR · 2023-09-18

**Recommendation:** 5

**Metareview:**

All the reviewers agree that the paper presents a well motivated approach, which is is also evaluated in a satisfactory manner and provides strong results. Not only did all reviewers agree on the soundness of the work, but also provided high excitement scores. This paper is an easy recommendation for the main conference.

However I would like to highly encourage the authors to pay attention to the recommendations of the reviewers to improve the presentation of the paper, as well as to the questions for clarification. Specifically, reviewer 4EMT pointed out that the paper may be hard to understand for people not completely familiar with WLAC. Please take the advice of all the reviewers into account when preparing the final submission.

---

### Decision · Program_Chairs · 2023-10-07

**Decision:**

Accept-Main

**Comment:**

All the reviewers agree that the paper presents a well motivated approach, which is is also evaluated in a satisfactory manner and provides strong results. Not only did all reviewers agree on the soundness of the work, but also provided high excitement scores. This paper is an easy recommendation for the main conference.

However I would like to highly encourage the authors to pay attention to the recommendations of the reviewers to improve the presentation of the paper, as well as to the questions for clarification. Specifically, reviewer 4EMT pointed out that the paper may be hard to understand for people not completely familiar with WLAC. Please take the advice of all the reviewers into account when preparing the final submission.